# Supervised learning using routine surveillance data improves outbreak detection of Salmonella and Campylobacter infections in Germany

**Benedikt Zacher** \*, **Irina Czogiel**

Department of Infectious Disease Epidemiology, Robert Koch Institute, Berlin, Germany

* zacherb@rki.de

## Abstract

The early detection of infectious disease outbreaks is a crucial task to protect population health. To this end, public health surveillance systems have been established to systematically collect and analyse infectious disease data. A variety of statistical tools are available, which detect potential outbreaks as abberations from an expected endemic level using these data. Here, we present supervised hidden Markov models for disease outbreak detection, which use reported outbreaks that are routinely collected in the German infectious disease surveillance system and have not been leveraged so far. This allows to directly integrate labeled outbreak data in a statistical time series model for outbreak detection. We evaluate our model using real Salmonella and Campylobacter data, as well as simulations. The proposed supervised learning approach performs substantially better than unsupervised learning and on par with or better than a state-of-the-art approach, which is applied in multiple European countries including Germany.

## Introduction

Infectious diseases are a significant threat to human health. Public health surveillance systems have been established to systematically collect, analyse and interpret data to guide public health actions [1]. A central part of public health surveillance is the early detection of disease outbreaks. Automated algorithms for outbreak detection are needed to handle the large amount of data that are collected [2]. The World Health Organization (WHO) defines a disease outbreak as "the occurrence of disease cases in excess of normal expectancy" [3]. Following the WHO definition, an algorithm for disease outbreak detection needs to address how to calculate the number of cases that is normally expected in a predefined geographical region and time window and define what an excess of cases is compared to the normal situation. The normal number of cases depends on the epidemiology of the considered disease. The number of cases might be sporadic or endemic in a certain region. Since we are only considering endemic diseases in this work, we further refer to the normally expected number of cases as the endemic level or baseline. The endemic level might further depend on seasonal or secular time trends,

**Data Availability Statement:** Case-based data of Salmonella and Campylobacter used in this study is collected based on the German Protection Against Infection Act infection act

(https://www.gesetze-im-internet.de/ifsg/) and is classified as pseudonymized personal data and therefore cannot be shared according to the General Data Protection regulation (https://gdpr.eu/). Aggregated data (case counts stratified by age, district and week of notification) can be accessed online (https://survstat.rki.de/). Additional aggregated data used for scientific purposes can be requested at RKI, but requires clearance from the RKI data protection officer. Requests can be sent to the corresponding author or info@rki.de.

**Funding:** BZ was supported by BMBF (Medical Informatics Initiative: HIGHmed) and the collaborative management platform for detection and analyses of (re-)emerging and foodborne outbreaks in Europe (COMPARE: European Union's Horizon research and innovation programme, grant agreement No. 643476).

**Competing interests:** The authors have declared that no competing interests exist.

that need to be accounted for. A plethora of methods are available to accomplish this task, most of which use either regression techniques or statistical process control (e.g. [4–8]). A review of a variety of methods can be found in [9]. In the following we will discuss two approaches in more detail. One approach is the use of a statistical background model to detect abberations from an expected endemic baseline. The second approach employs a model which explicitly includes an oubtreak state in addition to the endemic baseline and assigns one these states (i.e. endemic or outbreak) to an observed number of cases.

We illustrate the first approach by giving a short overview of the Farringtion-Noufaily (FN) algorithm, which is a widely used method for disease outbreak detection. The FN algorithm is an improved version of the original Farrington algorithm, which is a regression-based method for disease outbreak detection, that has been used in multiple European countries [5, 7, 10]. The method takes as input a time series which reflects the incidence of a disease for a given region along time. Throughout the manuscript we consider time series which are aggregated by week and county—i.e. each time series gives the number of weekly new cases for a county. However, other aggregation levels such as grouping multiple counties or regions together or stratifications by age groups are possible. The FN algorithm fits a quasi-Poisson Generalised Linear Model (GLM) of the endemic baseline on past data, usually five years, accounting for possible time trends and seasonal patterns. Case counts in the past that deviate by a predefined threshold from the fitted model are downweighted to avoid that the estimated enedemic baseline is baised by potential past outbreaks. Moreover, the most recent weeks are excluded to avoid an influence of recent outbreaks. Then the model predicts the endemic baseline for the current week and calculates an alarm threshold either based on a prediction interval or a quantile of the negative Binomial distribution to account for overdispersion in the data. If the number of cases in the current week exceeds the threshold, an alarm report is created for further investigation.

The second approach is illustrated at the example of hidden Markov models (HMMs), which also have been applied previously in infectious disease surveillance and outbreak detection [11–14]. HMMs are statistical models which assume that the data is generated by a sequence of underlying states. In the context of outbreak detection, the data is modeled by an endemic and an outbreak state. The endemic state generates data that refelcts the expected endemic baseline and the outbreak state generates data that reflect the expected number of cases in an outbreak. HMMs can be combined with GLMs, e.g. with the GLM used by the FN algorithm, to control for seasonal and secular time trends. After model fitting the HMM calculates a probability that the observed number of cases in the current week was generated from the outbreak state. If the probability of an outbreak exceeds a threshold, an alarm report is generated. A detailed formal definition and description of HMMs can be found in the Methods section.

HMMs can be fitted in an unsupervised and supervised manner. In the case of unsupervised model fitting the sequence of (past) states is unknown. This type of model fitting is also done with the FN algorithm. If HMMs are fitted in a supervised manner, the (past) sequence of states that generated the data are known. This means that the HMM can be trained on known instances of past endemic and outbreak states.

In Germany, data about notifiable infectious diseases is continuously reported from local and federal health authorities to the Robert Koch Institute (RKI) and transmitted using an electronic surveillance system (SurvNet), which was established in 2001 [15]. Besides the information about cases, disease outbreaks that are detected and confirmed e.g. by local health authorities are also reported according to the German 'Protection against Infection Act' [16]. The reported outbreaks collected in the SurvNet database have not been leveraged for outbreak detection so far. The aim of this study is the integration of this routinely collected data into an

HMM-based outbreak detection algorithm. We assess the benefit of using this information by comparing supervised HMMs, unsupervised HMMs and the FN algorithm.

## Materials and methods

We assume that our data are governed by a HMM. A HMM consists of a sequence of hidden states which generate the observed data. The evolution of hidden states is modeled by a first-order Markov chain which means that the hidden state at timepoint $t$ only depends on the hidden state at the previous time point $t - 1$. In our application, the observed data are weekly case counts of a disease in a predefined region. This data is generated by two unobserved (hidden) states, the endemic and the outbreak state. Here, unobserved or hidden means that we do not know whether there is an outbreak or not at any time point. The endemic state assumes weekly cases that are expected under normal conditions. The outbreak state assumes an increased incidence of cases compared to the endemic state. Our goal is to fit a model on past data, which is then used to calculate the probability of an outbreak in the current week. A (multiplicative) factor by which the incidence increases in the outbreak state is learned from the data. Since outbreaks are rare events, the HMM is trained on multiple time series in order to make sure that enough training examples for the outbreak state are available. We now give a more formal definition of the HMM.

We specify our model using a set of $N$ time series. For the sake of clarity we assume that all time series have the same length $T$, but the model can be applied to time series with different lengths. For any time point $t \in [1; T]$ in time series $n \in [1; N]$, the corresponding obervation $o_{n,t}$ is emitted (generated) from a hidden state $s_{n,t} \in \mathcal{K}$ that evolves over time according to a first-order Markov process. Our HMM has the following components:

- A set of $\mathcal{K} = \{0, 1\}$ states, where 0 represents the endemic and 1 the outbreak state.

- $\mathcal{O} = \{O_1, \ldots, O_N\}$ are the sequences of observations, where $O_n = (o_{n,1}, \ldots, o_{n,T})$ is the obervation sequence of a single time series. Each observation sequence gives the number of cases aggregated by a predefined time period (e.g. weekly) and region.

- $\mathcal{S} = \{S_1, \ldots, S_N\}$ gives the corresponding sequences of hidden states, where $S_n = (s_{n,1}, \ldots, s_{n,T})$ and $s_{n,t} \in \mathcal{K}$.

- $\pi_i = \Pr(s_{n,1} = i)$ is the vector of initial state probabilities, where $\sum_{j \in \mathcal{K}} \pi_j = 1$. This reflects the probability of the endemic or outbreak state at the first time point of the time series.

- $a_{ij} = \Pr(s_{n,t} = j | s_{n,t-1} = i)$ are transition probabilities between the states $i, j \in \mathcal{K}$, with $\sum_{j \in \mathcal{K}} a_{ij} = 1$. For instance $a_{01}$ gives the probability that timepoint $t$ is in the outbreak state given that the previous time point $t - 1$ is in the endemic state.

- $\psi_{s_{n,t}}(o_{n,t})$ is a vector of emission functions $\psi_{s_{n,t}}(o_{n,t}) = \Pr(o_{n,t} | s_{n,t},)$, which calculate the probability of observing $o_{n,t}$ cases given state $s_{n,t}$.

- The parameter vector $\theta = (\pi_i, a_{ij}, \psi)$ specifies the HMM.

As mentioned above, the observations are the reported number of cases of a disease during a certain time, e.g. weeks. Hidden state $s_{n,t} = 1$ indicates an ongoing outbreak with an excess number of cases in week $t$, and $s_{n,t} = 0$ applies to weeks where the case number is consistent with the expected baseline (endemic). Many infectious diseases and hence the corresponding surveillance data follow an annual pattern and a general time trend—i.e. overall decrease or increase of the number of cases over time. Thus the probability of the number of cases—$\Pr(o_{n,t} | s_{n,t})$—depends on time and the season of the year. We employ the

Farrington-Noufaily model which uses a generalised linear model to adjust for these secular and seasonal trends [7]. The FN model estimates the expected number of cases in the endemic state ($s_{n,t} = 0$) by incorporating a linear time trend and uses a 10-level factor to represent different time periods throughout a year to account for seasonal patterns. We extend this model by introducing an additional parameter that captures the increase of the expected number of cases in the outbreak state $s_{n,t} = 1$. The log of the expected number of cases $\mu_{n,t}$ in time series $n$ at time $t$ is given by

$$\log \mu_{n,t} = \beta_{n,intercept} + \beta_{n,time}t + \beta_{n,season(t)} + \beta_{outbreak}s_{n,t}$$

Here, $\beta_{n,intercept}$ is the intercept, $\beta_{n,time}$ a secular time trend and $\beta_{n,season(t)}$ describes seasonal patterns, where $season(t)$ is a function indicating which season or period of the year $t$ belongs to. The state-dependence of $o_{n,t}$ on $s_{n,t}$—and thus the effect of an outbreak—is incorporated by a multiplicative factor $\exp(\beta_{outbreak})$ that models the excess number of cases in outbreak situations. Note that, while $\beta_{n,intercept}, \beta_{n,time}, \beta_{n,season(t)}$ are distinct for each time series $n \in [1; N]$, $\beta_{outbreak}$ is the same for all. This allows information about past outbreaks to be shared across time series. Note that the model assumes that the increase in case numbers in an outbreak is independent of the population size. Considering multiple time series at once during model fitting is necessary to make the model robust, since the number of outbreaks can vary greatly between time series. If only a few outbreaks occurred in the training data of a single time series, fitting a model with a specific parameter for the effect of an outbreak would not generalize well on new data. In particular, it would not be possible to fit a model on a single time series if there is no outbreak in the training data.

We assume that the data in our surveillance time series follows a negative Binomial distribution: $o_{n,t} \sim \text{NB}(\mu_{n,t}, r_n)$, where $r_n$ is the dispersion and $\mu_{n,t}$ the mean of the negative Binomial distribution. The negative Binomial distribution is widely used to model surveillance data and allows to account for overdispersion. When $r \rightarrow \infty$ with fixed mean the Negative Binomial reduces to a Poisson distribution. We conducted additional analyses using a Poisson model, i.e. $o_{n,t} \sim \text{Pois}(\mu_{n,t})$, to compare the performance between the two models.

## Parameter learning

Since there might be a reporting delay of outbreaks—i.e. outbreaks in the recent past are not yet recorded in SurvNet—we exclude $u$ time units from our training data: $T_{train} = T - u$. Let $S_n^{train} = \left(s_{n,1}, \ldots, s_{n,T_{train}}\right) \in \mathcal{S}_{train}$ and $O_n^{train} = \left(o_{n,1}, \ldots, o_{n,T_{train}}\right) \in \mathcal{O}_{train}$ denote the known state and observation sequences used for model training. Assuming independence between individual time series, the likelihood of our model is:

$$\Pr(\mathcal{O}_{train}, \mathcal{S}_{train}|\theta) = \prod_{n=1}^{N} \Pr(O_n^{train} \mid S_n^{train}, \theta) \cdot \Pr(S_n^{train}|\theta)$$

$$= \prod_{n=1}^{N} \left[ \prod_{t=1}^{T_{train}} \psi_{s_{n,t}}(o_{n,t}) \cdot \prod_{t=1}^{T_{train}} a_{s_{n,t-1},s_{n,t}} \cdot \pi_{s_{n,0}} \right]$$

Maximum likelihood estimation of model parameters $\pi_i$ and $a_{i,j}$ is straightforward:

$$a_{i,j} = \frac{\sum_{n=1}^{N}\sum_{t=2}^{T_{train}} \delta(s_{n,t-1},i)\delta(s_{n,t},j)}{\sum_{n=1}^{N}\sum_{t=2}^{T_{train}} \delta(s_{n,t-1},i)}$$

$$\pi_i = \frac{\sum_{n=1}^{N} \delta(s_1,i)}{N}$$

where $\delta(i,j) = \begin{cases} 1, & \text{if } i = j \\ 0, & \text{if } i \neq j \end{cases}$.. Both estimators are simply the observed relative frequencies in the training data, i.e. the observed relative frequency of transitions between state $i$ and $j$ and the observed relative frequency of state $i$ at the first time point.

Estimation of $\beta = (\beta_{n,intercept}, \beta_{n,time}, \beta_{n,season(t)}, \beta_{outbreak})$ and optimisation of the dispersion parameter $r$ is carried out using the Iteratively Reweighted Least Squares algorithm for Generalised Linear Models, for which the R functions glm.fit() and glm.nb() are used [17, 18].

## Posterior probability of an outbreak

The last time point $T$ (i.e. the current time point) of a time series is the time point of interest for outbreak detection. In order to determine whether time point $T$ in time series $n$ is in the endemic or in the outbreak state, the posterior probability

$$\Pr(s_{n,T} = 1|O_n, \theta) = \frac{\Pr(s_{n,T} = 1, O_n|\theta)}{\sum_{s_{n,T} \in \{0,1\}} \Pr(s_{n,T}, O_n|\theta)}$$

is calculated. This can be done efficiently using recursive computation of the forward probabilites $\alpha_{n,t}(i) = \Pr(s_{n,t} = i, O_n|\theta)$ of a HMM [11].

## Data and model fitting

Time series data were extracted for Salmonella and Campylobacter infections from the SurvNet database (accessible online: https://survstat.rki.de/), which collects reports about notifiable diseases at the RKI. Data were aggregated by disease, local health authorities—representing counties or districts in Germany—and weeks. Weekly outbreak labels were assigned to counties if at least two cases were part of a reported outbreak in that week, according to the Protection against Infection Act [16]. Time series were randomly assigned to 20 equally sized groups, ensuring that each group had enough outbreaks for training. For each week in 2010–2017 models were trained on data using the past five years excluding the 26 latest weeks. Models were fit in an unsupervised and supervised manner.

## Simulations

To assess model performance in a controlled setting, we adapted 14 different simulation scenarios as proposed in Noufaily et al. [7]. In short, expected case counts for each time series with a length of 624 were simulated from a linear model including Fourier terms for an annual seasonal pattern and an optional time trend: $\mu_t = \exp(\beta_0 + \beta_1 t + \beta_2 \cos(\frac{2\pi}{52} t) + \beta_3 \sin(\frac{2\pi}{52} t))$. Parameters for all simulation scenarios are depicted in S1 Table. The state sequences of endemic and outbreak weeks were simulated using a transition matrix, where for each time series, the probability of remaining in the endemic state, $a_{00}$, was sampled from a uniform distribution from the interval [0.9; 1] and the probability of remaining in the outbreak state, $a_{11}$, was sampled from [0.4; 0.6]. As an additional parameter, each scenario is assigned a dispersion parameter $\phi \geq 1$ (S1 Table). Weekly case counts of the endemic state were then sampled from a negative Binomial with mean $\mu_t$ and variance $\phi\mu_t$. The number of cases in outbreaks was drawn from a Poisson distribution with mean $2 \cdot \sqrt{\phi\mu_t}$. The sampled number of cases in the oubtreak was added to the sampled endemic level. To make sure that simulated outbreaks had at least two cases we added 2 to the number cases sampled.

### Evaluation and benchmarking

Performance of the supervised and unsupervised HMM was compared to the FN algorithm, which is currently the method of choice for outbreak detection at the RKI. The farringtonFlexible() function from the R package surveillance was used with the following (default) control parameters: noPeriods = 10, b = 5, w = 3, weightsThreshold = 2.58, pastWeeksNotIncluded = 26, thresholdMethod = 'nbPlugin', alpha = 0.01 [19]. ROC curves were computed using the R package ROCR [20]. We calculated the sensitivity (recall) of the methods using two approaches. The first approach simply calculates the fraction of individual outbreak weeks that were recalled. However, if outbreaks span more than one one week, true positives are counted mutliple times if the method triggers an alarm at multiple weeks during the outbreak. Therefore, we use a second measure which calculates the recall of outbreaks such that outbreaks spanning more than one week are only counted once. We refer to these measures as 'recall of outbreak weeks' and 'recall of outbreaks'. Moreover, since many outbreaks are small, we calculated ROC curves using (i) all outbreaks and (ii) outbreaks with at least four cases in the outbreak.

### Availability

The R source code for fitting the supervised HMM is available in S1 File. Data of Salmonella and Campylobacter infection can be accessed online (https://survstat.rki.de/), however due to legal restrictions, the outbreak data cannot be made available publicly.

## Results

We applied the HMM with the negative Binomial distribution to simulated and real infectious disease data. First we illustrate and benchmark the method using simulated data, followed by an analysis using real Salmonella and Campylobacter data. To compare the negative Binomial with the Poisson distribution, we also evaluated the performance of the HMM using the Poisson distribution.

We start with a brief explanation and illustration of the workflow and components of our method using simulated data (Fig 1). The Figure shows application to twelve simulated time series. For illustration, these were mapped to the twelve districts of Berlin. The HMM uses the past five years of the twelve time series as training data, excluding the past 26 weeks to avoid model training on incomplete data due to possible reporting delays. Since disease outbreaks are rare events, the model is trained on multiple (in this case twelve) time series at once to make sure that enough training examples for the outbreak state are available (Fig 1A). The fitted HMM consists of a linear predictor, initial state and transition probabilities (Fig 1B). The linear predictor is used to estimate the expected number of cases for the endemic and the outbreak state. This estimate is used together with the initial state and transition probabilities to calculate the probability of an outbreak in the current week (Fig 1C). An alarm is be triggered if the probability exceeds a chosen threshold.

Performance of the supervised and unsupervised negative Binomial HMMs were compared to the FN algorithm using 14 simulation scenarios (Fig 2). Two examples of simulated time series are shown in Fig 2A and 2B, illustrating the seasonal and general time trends that are often observed in real infectious disease surveillance data. Considering the recall of outbreak weeks, the supervised HMM outperforms the unsupervised HMM and the FN algorithm (Fig 2C). The supervised HMM and the FN algorithm perform similar when considering the recall of outbreaks and both perform better than the usupervised HMM (Fig 2D). Note that the recall was calculated for oubtreak weeks and additionally for distinct outbreaks to avoid counting multiple weeks of one outbreak as true positives (see Methods for details). Known outbreak

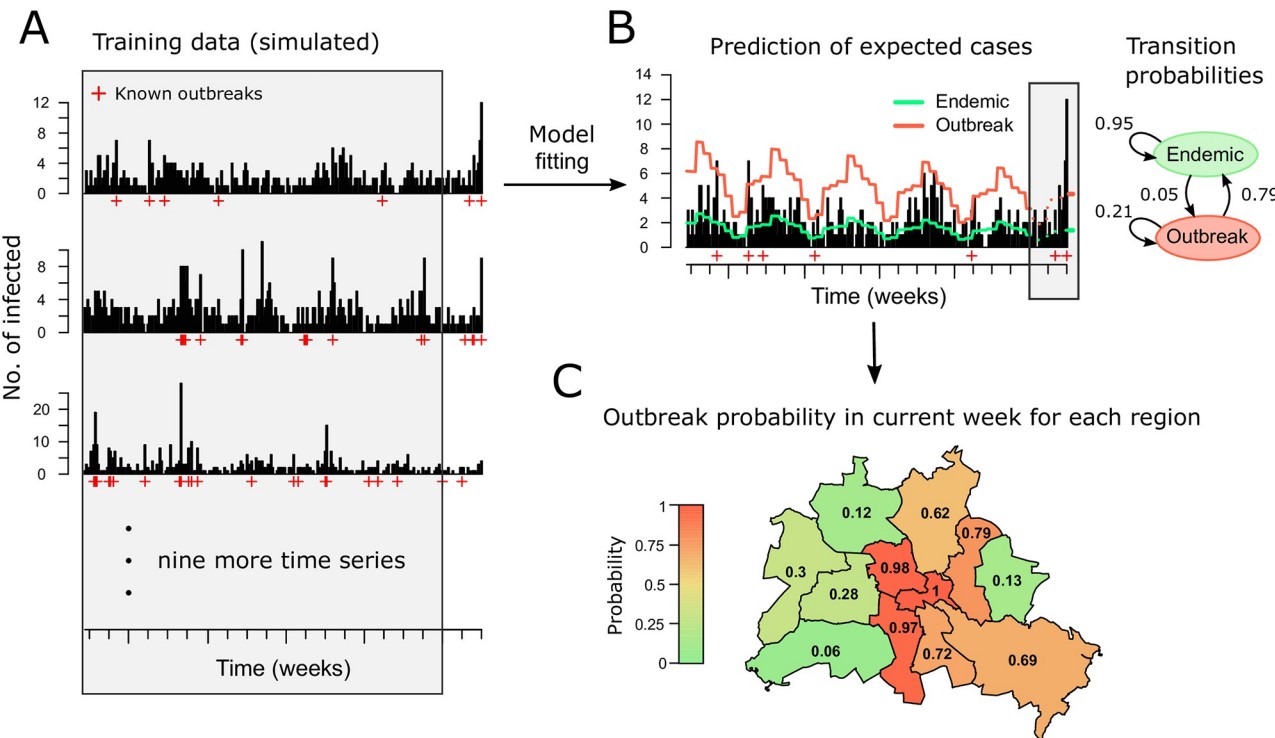

**Fig 1. Overview of the method using simulated data.** For illustrative purposes the data was assigned to the twelve districts of Berlin. We use simulated data, because outbreak data cannot be shown for individual counties due to legal restrictions. To get a realistic scenario for illustration of the method, the data was simulated from HMMs which were fit to real data of Salmonella infections. (A) The example shows five years of data simulated of for multiple counties. Training data is indicated by the shaded area. (B) The expected number of cases in the endemic (green) and the outbreak (red) state are shown, which are extrapolated up to the current week (dashed lines). The fitted transition proabablities are shown as a graph. (C) For each region, the outbreak probabilities are calculated.

and endemic weeks are well separated by the outbreak probability calculated by the HMM across all simulation scenarios (Fig 2E). The HMM using the Poisson distribution performed worse than the HMM using the negative Binomial distribution on the simulated data (S1 Fig).

We also applied the HMM and the FN algorithm to real Salmonella and Campylobacter data from more than 400 counties in Germany. Fig 3 shows Salmonella and Campylobacter data aggregated by week for Germany. The number of infections and outbreaks per week in Germany show a strong seasonal pattern and there is a decrease of Salmonella cases and a low increase of Campylobacter cases over time. This justifies the choice of our model to include seasonal and secular trends. We applied the models to detect outbreaks from 2010 to 2017. During this time there were 2,124 Salmonella outbreaks with a duration of 1–8 weeks and 2,259 Campylobacter outbreaks with a duration of 1–16 weeks. The average weekly number of cases in counties which reported an outbreak exhibits a marked increase compared to the average cases in counties where no outbreak was reported.

The supervised negative Binomial HMM outperformed the unsupervised HMM and the FN algorithm on the Salmonella and Campylobacter data (Fig 4). All methods showed a relatively low recall when all outbreaks were considered for evaluation (Fig 4A, 4B, 4D and 4E). The recall was significantly higher, when calculated for larger outbreaks involving at least four cases overall (recall of outbreaks) or in a week (recall of outbreak weeks). This is in accordance with the increase in HMM outbreak probabilities with the size of reported outbreaks (Fig 4C and 4F). Outbreak probabilites in reported outbreak weeks were generally higher for

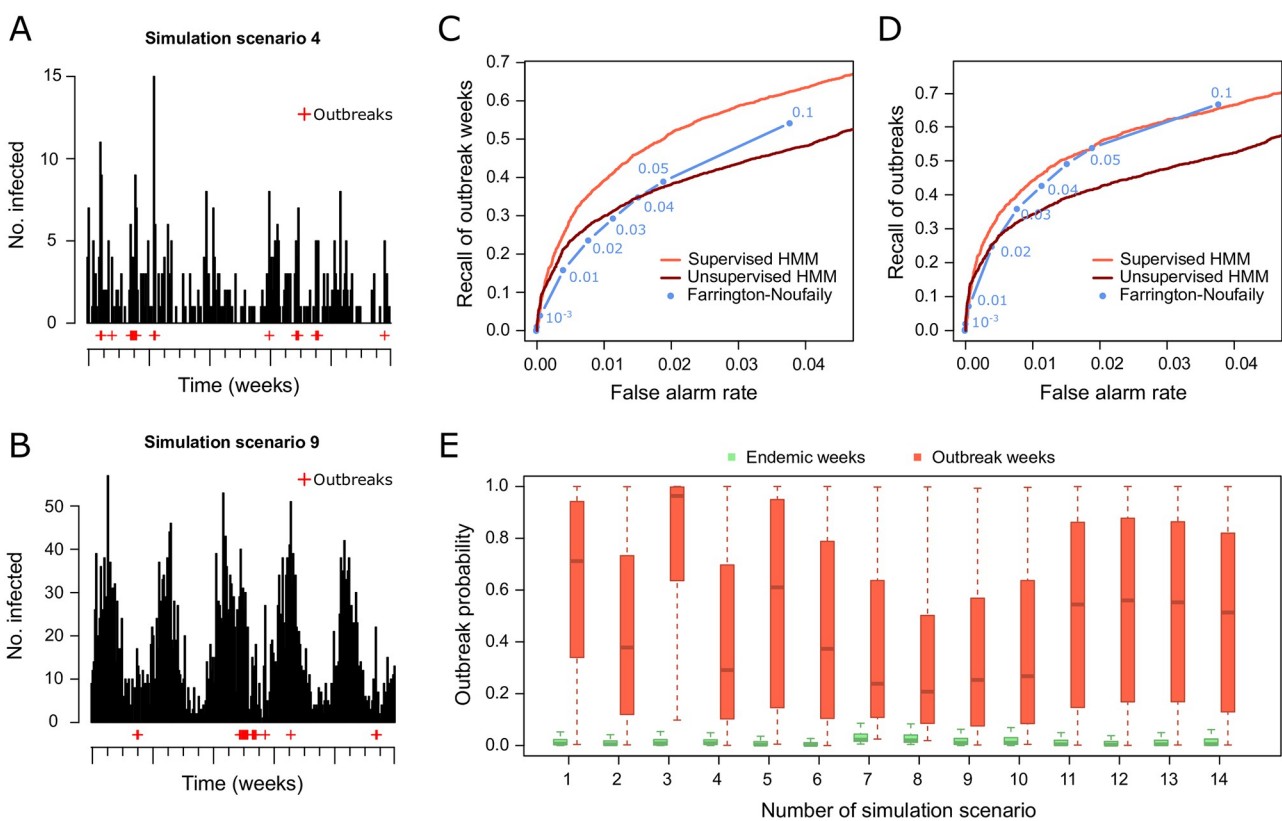

**Fig 2. Benchmark of the negative Binomial hidden Markov model (HMM) and the Farrington-Noufaily algorithm on simulated data set.** (A) Example simulated time series for scenario 4. (B) Example simulated time seires for scenario 9. (C) ROC curve showing the false alarm rate and recall of outbreak weeks for all 14 simulation scenarios. (D) ROC curve showing the false alarm rate and recall of outbreaks for all 14 simulation scenarios. (E) Boxplots of posterior probabilities of known endemic (green) and outbreak (red) weeks for all 14 simulation scenarios.

Salmonella than Campylobacter data, i.e. outbreak weeks can be better distinguished from endemic weeks in the Salmonella than in the Campylobacter data. This matches the fitted outbreak effects ($\exp(\beta_{outbreak})$, see Methods) of the HMMs. The average increase in the number of cases during an outbreak ranged from 1.9—7.7 (mean: 3.3) for Salmonella and 1.2—2.7 (mean: 1.8) for Campylobacter. Performance of the supervised Poisson HMM was slightly better than for the negative Binomial HMM (S2 Fig). Interestingly, the unsupervised Poisson HMM performed significantly better than the unsupervised negative Binomial HMM. Compared to the FN algorithm, false alarm rate and recall were similar on the Salmonella data and favorable on the Campylobacter data.

To investigate whether there are differences between the outbreaks recalled by the HMM and the FN algorithm, we calculated the overlap of the alarms of both methods and with known outbreaks from the SurvNet database for the Salmonella and Campylobacter data (Fig 5). The alarm threshold for the HMM was chosen such that the recall of the HMM was the same as the recall of the FN algorithm with a cutoff of 0.01 using threshold method 'nbPlugin'. Despite a large overlap of correctly recalled outbreaks between methods, we found that 17% (Salmonella) and 32% (Campylobacter) of recalled outbreaks were only recalled by one of the methods (Fig 5A and 5D). Further analyses showed that outbreaks that were recalled by both methods were larger in size than oubtreaks that were recalled by only one method or missed by both (Fig 5B and 5E). Moreover, outbreaks that were only recalled by the HMM

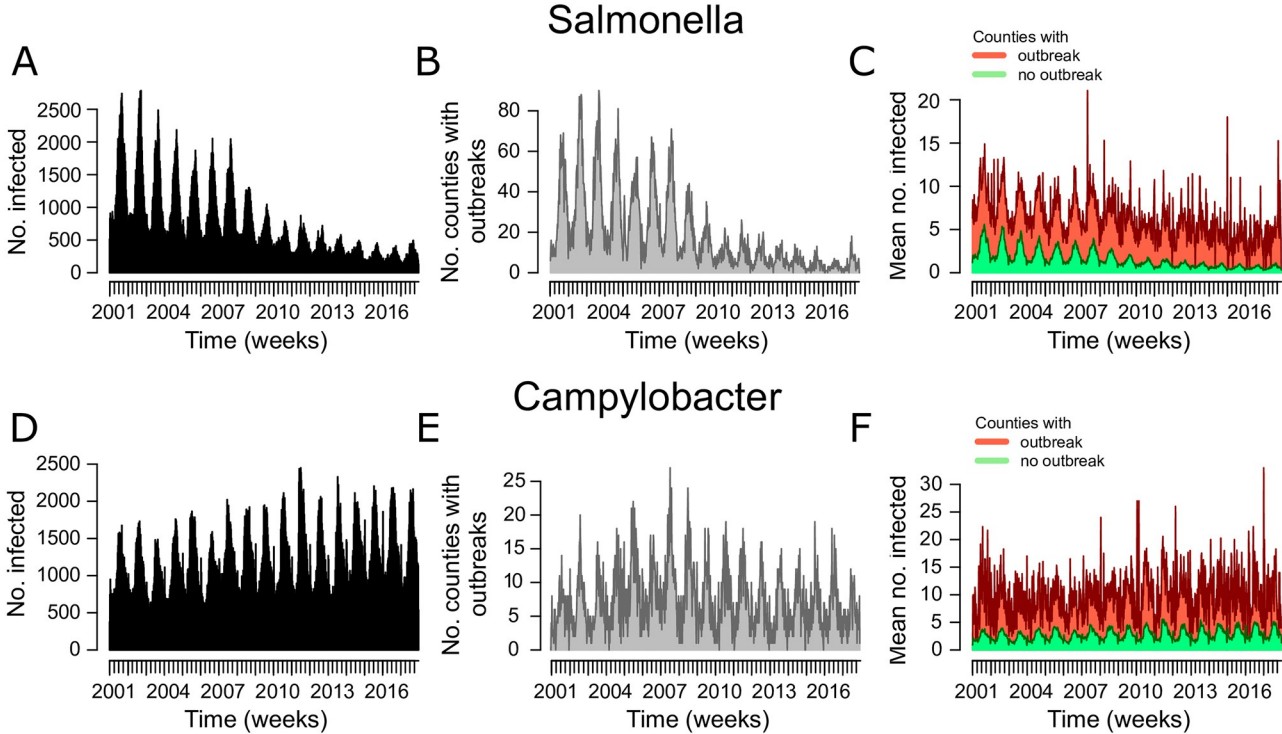

**Fig 3. Aggregated weekly data of Salmonella and Campylobacter infections in Germany reported to the Robert Koch Institute.** (A) The number of Salmonella infections. (B) The number of counties with a reported Salmonella outbreak. (C) The mean number of cases in counties with a reported Salmonella outbreak (red) and the mean number of cases in counties without outbreak (green). (D-F) The same as (A-C) for Campylobacter infections.

were larger than those that were only recalled by the FN algorithm. A comparison of outbreak durations showed that outbreaks recalled by both methods were longer than those recalled by only one of the methods (Fig 5C and 5F). 43% of these outbreaks had a duration of two weeks or more. 35% of the outbreaks only recalled by the HMM—but not the FN algorithm—and 15% of the outbreaks only recalled by the FN algorithm—but not the HMM—were at least two weeks long. Outbreaks that were missed by both methods were shorter with only 8% having a duration of two or more weeks. These results were consistent with the results obtained for the Poisson HMM (S3 Fig).

## Discussion

We developed supervised hidden Markov models for outbreak detection using routine surveillance data. Infectious disease surveillance data in Germany contains information about known and reported outbreaks which has not used for this task so far. This study integrates these data in a statistical time series model for outbreak detection. Application of the supervised HMM showed its potential to improve performance of outbreak detection on simulated and real data.

Our analyses showed that supervised learning performed better than unsupervised learning. This is not surprising since supervised learning has access to labeled data during model training, while the unsupervised learning does not have access to this information. However, this result indicates that the use of routinely reported outbreaks can improve outbreak detection algorithms. We also compared HMMs using the negative Binomial and the Poisson distribution. Both distributions performed well on the real data sets with supervised learning.

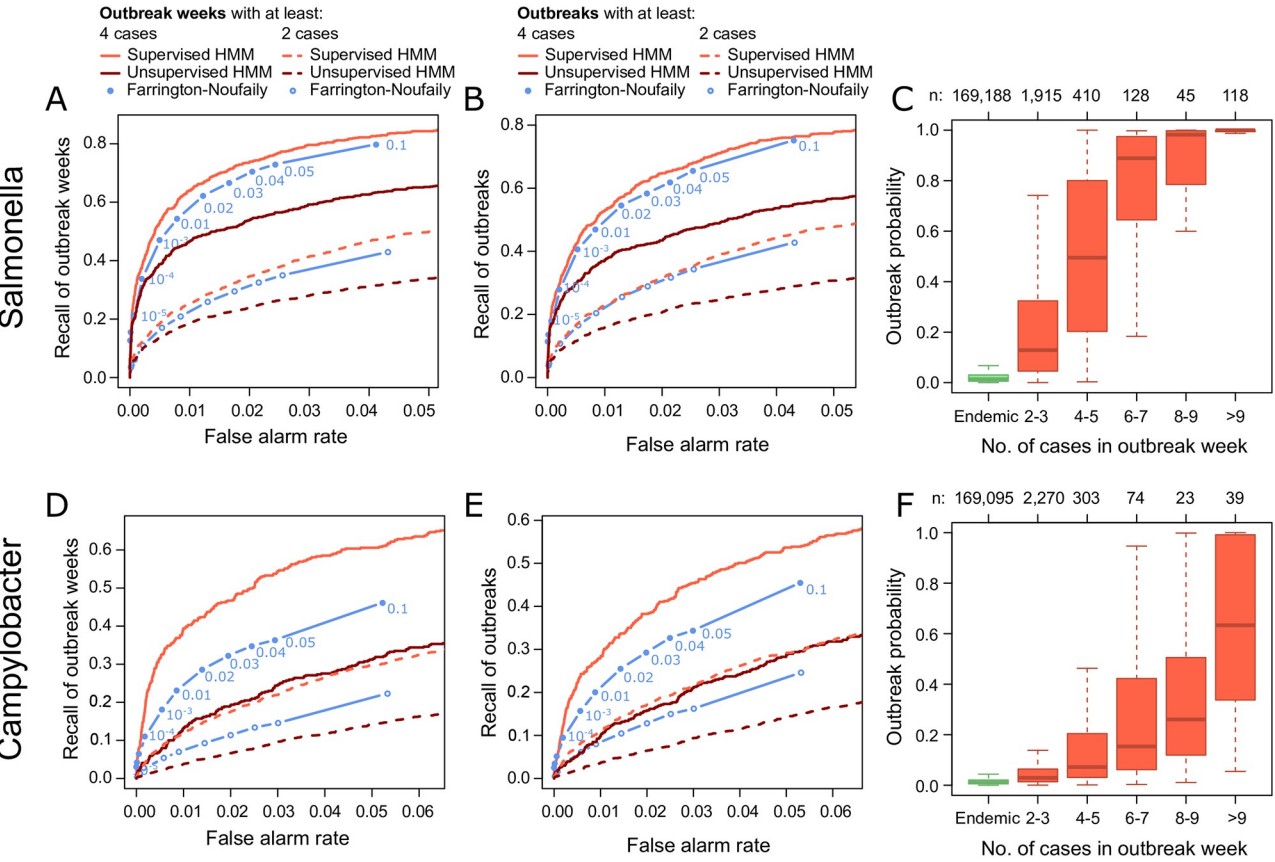

**Fig 4. Benchmark of the negative Binomial hidden Markov model (HMM) and the Farrington-Noufaily (FN) algorithm on real Salmonella and Campylobacter data from 2010–2017.** (A) ROC curve showing the false alarm rate and recall of oubtreak weeks for the Salmonella data. Performance was evaluated with outbreaks that involved at least two or four cases. The FN alogrithm was applied with cutoffs $10^{-6}$, $10^{-5}$, $10^{-4}$, 0.001, 0.005 and 0.01 using threshold method 'nbPlugin'. (B) The same as (A) but showing the recall of oubtreaks. (C) Boxplots of oubtreak probabilities form the HMM are shown for known endemic (green) and outbreak (red) weeks. Outbreaks were further divided by their size (i.e. the number of cases reported in an outbreak per week). (D-F) The same as (A-C) for Camplyobacter data.

Interestingly, the unsupervised Poisson HMM performed much better than the unsupervised negative Binomial HMM on the Salmonella and Campylobacter data sets. Here, the additional free parameter in the negative Binomial distribution to account for overdispersion seems to complicate model fitting when no labeled data is available. On the simulated data, the Poisson distribution performed worse than the negative Binomial distribution This might be due to the overdispersion that was integrated in most of the simulation scenarios (S1 Table). Thus, if labeled training data is available, the supervised negative Binomial might be a good choice that performs well accross different scenarios. If model training is conducted in an unsupervised manner, both distributions should be applied and compared to pick the best model.

The supervised negative Binomial HMM performed on par with or better than the FN algorithm on simulated and better on real Salmonella and Campylobacter data. Outbreaks that were recalled by the HMM tended to be longer and larger than those recalled by the FN alogrithm. The HMM models the evolution of endemic and outbreak states over time, while the FN algorithm does not consider such dependencies. For instance, consider an oubtreak spanning several weeks, but each week only shows a slight increase of cases compared to the expected endemic level such that the alarm threshold is not exceeded in the first week of the

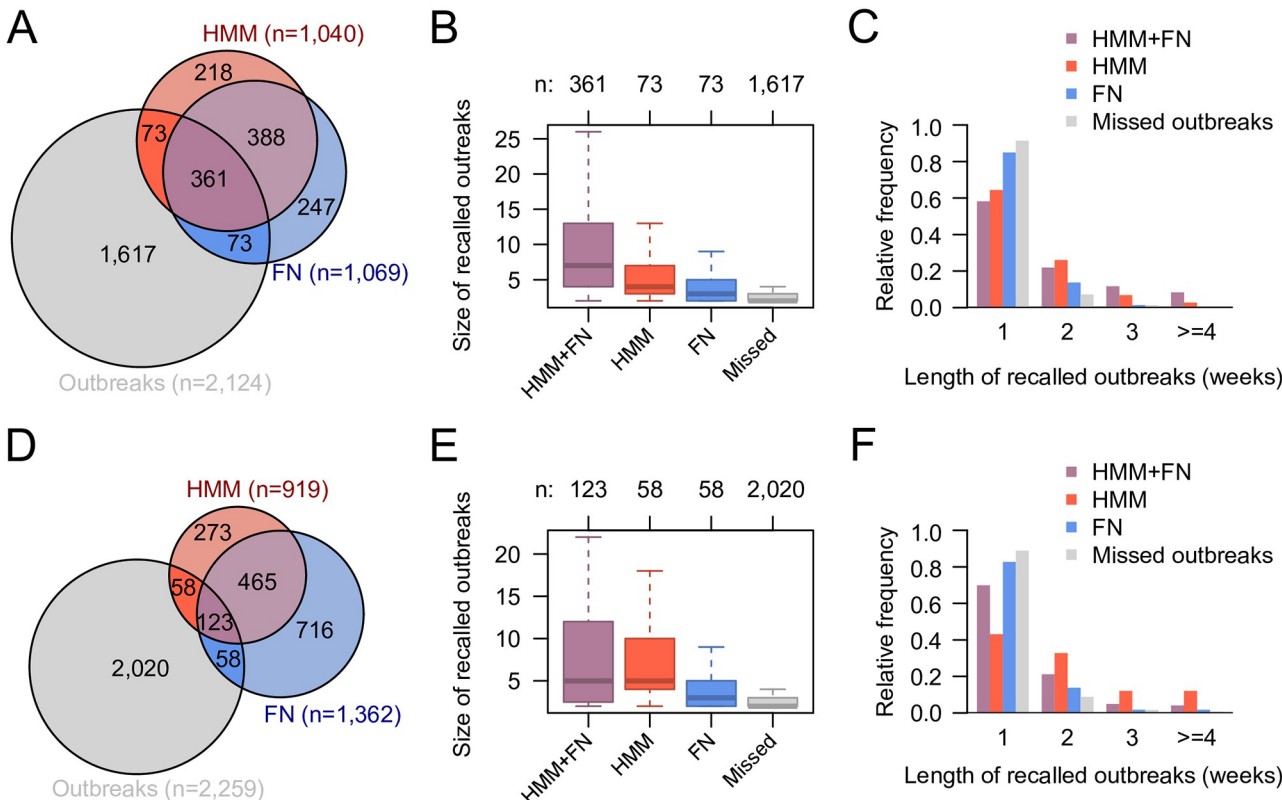

**Fig 5. Comparison of outbreaks recalled by the supervised negative Binomial hidden Markov model (HMM) and the Farrington-Noufaily (FN) algorithm.** (A) Venn diagram showing the overlap between outbreaks recalled by both methods. The FN algorithm was applied with cutoff 0.01 using threshold method 'nbPlugin'. The cutoff for the HMM outbreak probability was chosen to get the same recall as the FN algorithm. (B) The number of cases in outbreaks recalled by both, one or none of the applied methods. (C) Distribution of the duration (in weeks) of outbreaks recalled by both, one or none of the applied methods. (D-F) The same as (A-C) for the Campylobacter data.

outbreak by both methods. In this setting the outbreak probability will increases over the course of the outbreak in the HMM and might exceed the alarm threshold at some point, since the evolution the states (endemic and outbreak) at previous time points is considered when calculating the outbreak probability. In contrast, the FN algorithm assumes independence between time points and its alarm threshold will not adjust as the outbreak goes along.

It is also important to discuss some limitations of our approach. One obvious caveat of any supervised learning approach is, that outbreak labels are needed for model training. This data might not be available in other surveillance systems or might not be collected for all diseases of interest. However in such cases one can resort to other well established algorithms such as the FN algorithm or use the unsupervised Poisson HMM which also performed well on the real data sets. The performance of a supervised learning approach for outbreak detection also depends on the quality of the outbreak labels. Our model exploits the fact that outbreak weeks exhibit an increase in case numbers compared to endemic weeks. However, weeks with an excess number of cases are not reported (and labeled) as outbreaks if the compulsory registration criteria defined in the Protection against Infection Act are not met. This is not problematic for our approach as long as reported outbreak weeks show an average increase of case numbers, separating outbreak from endemic weeks. This is the case for average case counts aggregated for weekly endemic and epidemic weeks for Germany (Fig 3). It is also verified by

the fitted models, since the 'outbreak effect' parameters $\exp(\beta_{outbreak})$ show a strong average increase in the number of cases in outbreak weeks.

An assumption of the presented model is that the increase in cases in outbreaks compared to the endemic state is multiplicative. Therefore, our model will not detect smaller outbreaks with increasing endemic case numbers. However, the relevance of small outbreaks for public health surveillance and infection control is also limited, if the background rates are high, because small outbreaks do not contribute much to the overall occurrence of infection.

Another issue is that most outbreaks that are reported are small. 73% (n = 1,915) of Salmonella and 83% (n = 2,270) of Campylobacter oubreaks only comprised 2–3 cases. These small outbreaks might not cause a substantial increase over the endemic level and are therefore difficult to detect. This explains the low sensitivity on the real data set when considering all outbreaks for evaluation (Fig 4). However, this is not problematic in practice because the public health threat that comes from these small outbreaks is limited. Moreover, the sensitivity increases substantially when considering outbreaks with at least four cases and thus larger and more severe outbreaks are likely detected.

Another limitation is that the assumption of (conditional) independence of observed time points in modeling infectious disease surveillance data is questionable, since the number of infections from one week might affect the next week. Although our proposed HMM does not take into account the dependence of subsequent observations, it incorporates dependence of the state of a current week (i.e. outbreak or endemic) on the previous week. Moreover, our model assumes that time series of individual counties are independent. This is likely not the cases, especially for neighbouring counties. However, these simplifying assumptions are justified by the good performance in the practical application to Salmonella and Campylobacter data.

The models were applied to time series aggregated by week since the vast majority of outbreaks in the considered datasets cover only one week (85% for Salmonella and 86% for Campylobacter). Applying the methods to larger aggregation levels such as 2 or 4 week periods would dilute the increased incidence of most outbreaks, since outbreak weeks would be aggregated with adjacent non-outbreak weeks. However, in principle other aggregation levels are possible.

The HMM was tested for two food-borne infectious diseases, but application is possible for other infectious diseases such as the detection of Influenza epidemics [13]. Application of the supervised HMM is possible as long as enough labeled outbreak data is available for model training. This is likely the case for diseases that exhibit a certain endemic baseline level. Supervised learning approaches are less suitable for sporadic diseases where only single or few cases occur with long intermittent intervals with no cases. Here, on one hand only very few outbreaks occur and thus few training examples are available. On the other hand, depending on the severity of the disease an alarm might already have to be triggered with only one occuring case. Thus health care workers and epidemiologists would be alarmed with occurence of a single case and in practice no automated methods for detection are needed.

In summary, our results are promising that leveraging outbreak data with supervised learning may improve disease outbreak detection. Thus we foresee our approach to be instrumental to improve public health surveillance systems in the future.

## Supporting information

**S1 Fig. Benchmark of the Poisson hidden Markov model (HMM) and the Farrington-Noufaily algorithm on simulated data set.** (A) ROC curve showing the false alarm rate and recall of oubtreak weeks for all 14 simulation scenarios. (B) ROC curve showing the false alarm rate

and recall of oubtreaks for all 14 simulation scenarios. (C) Boxplots of posterior probabilities of known endemic (green) and outbreak (red) weeks for all 14 simulation scenarios.
(PDF)

**S2 Fig. Benchmark of the Poisson hidden Markov model (HMM) and the Farrington-Noufaily (FN) algorithm on real Salmonella and Campylobacter data from 2010-2017.** (A) ROC curve showing the false alarm rate and recall of oubtreak weeks for the Salmonella data. Performance was evaluated with outbreaks that involved at least two or four cases. The FN algorithm was applied with cutoffs $10^{-6}$, $10^{-5}$, $10^{-4}$, 0.001, 0.005 and 0.01 using threshold method 'nbPlugin'. The HMM was applied with the negative Binomial distribution. (B) The same as (A) but showing the recall of oubtreaks. (C) Boxplots of oubtreak probabilities form the HMM are shown for known endemic (green) and outbreak (red) weeks. Outbreaks were further divided by their size (i.e. the number of cases reported in an outbreak per week). (D-F) The same as (A-C) for Camplyobacter data.
(PDF)

**S3 Fig. Comparison of outbreaks recalled by the supervised Poisson hidden Markov model (HMM) and the Farrington-Noufaily (FN) algorithm.** (A) Venn diagram showing the overlap between outbreaks recalled by both methods. TheFN algorithm was applied with cutoff 0.01 using threshold method 'nbPlugin'. The cutoff for the HMM outbreak probability was chosen to get the same recall as the FN algorithm. (B) The number of cases in outbreaks recalled by both, one or none of the applied methods. (C) Distribution of the duration (in weeks) of outbreaks recalled by both, one or none of the applied methods. (D-F) The same as (A-C) for the Campylobacter data.
(PDF)

**S1 File. Supervised HMM implementation.** R source code of the supervised HMM.
(R)

**S1 Table. Simulations scenarios.** Parameters for all 14 simulation scenarios are shown.
(DOCX)

## Acknowledgments

We thank Alexander Ullrich for help with data extraction, preprocessing and feedback on the manuscript.

## Author Contributions

**Conceptualization:** Benedikt Zacher, Irina Czogiel.

**Investigation:** Benedikt Zacher.

**Methodology:** Benedikt Zacher.

**Supervision:** Benedikt Zacher.

**Writing – review & editing:** Irina Czogiel.

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
