## [Editor Report · Decision Letter 0]

10 Jan 2022

PONE-D-21-39547

Supervised learning using routine surveillance data improves outbreak detection of Salmonella and Campylobacter infections in Germany

PLOS ONE

Dear Dr. Zacher,

Thank you for submitting your manuscript to PLOS ONE. After careful consideration, we have decided that your manuscript does not meet our criteria for publication and must therefore be rejected.

Specifically:

ACADEMIC EDITOR: In the current paper, authors have presented supervised hidden Markov models for disease outbreak detection, which use reported outbreaks that are routinely collected in the German infectious disease surveillance system and have not been leveraged so far. This allows to directly integrate labeled outbreak data in a statistical time series model for outbreak detection. The novelty of the paper is limited and is not sufficient to be published in PLOS ONE journal. Hence I reject the paper.

I am sorry that we cannot be more positive on this occasion, but hope that you appreciate the reasons for this decision.

Yours sincerely,

Sriparna Saha, PhD

Academic Editor

PLOS ONE

Additional Editor Comments:

In the current paper, authors have presented supervised hidden Markov models for disease outbreak detection, which use reported outbreaks that are routinely collected in the German infectious disease surveillance system and have not been leveraged so far. This allows to directly integrate labeled outbreak data in a statistical time series model for outbreak detection. The novelty of the paper is limited and is not sufficient to be published in PLOS ONE journal. Hence I reject the paper.
---

## [Author Response · Author response to Decision Letter 0]

3 Feb 2022

The initial submission was rejetced prior to peer review. After appealing to this decsion, Plos One decided to reconsider the manuscript.

---

## [Decision Letter · Decision Letter 1]

14 Mar 2022

PONE-D-21-39547R1Supervised learning using routine surveillance data improves outbreak detection of Salmonella and Campylobacter infections in GermanyPLOS ONE

Dear Dr. Zacher,

Thank you for submitting your manuscript to PLOS ONE. After careful consideration, we feel that it has merit but does not fully meet PLOS ONE’s publication criteria as it currently stands. Therefore, we invite you to submit a revised version of the manuscript that addresses the points raised during the review process.

ACADEMIC EDITOR: Please insert comments here and delete this placeholder text when finished. Be sure to:Indicate which changes you require for acceptance versus which changes you recommendAddress any conflicts between the reviews so that it's clear which advice the authors should followProvide specific feedback from your evaluation of the manuscriptPlease ensure that your decision is justified on PLOS ONE’s publication criteria and not, for example, on novelty or perceived impact.

We look forward to receiving your revised manuscript.

Kind regards,

Hong Qin

Academic Editor

PLOS ONE

Journal Requirements:

1. Please ensure that your manuscript meets PLOS ONE's style requirements, including those for file naming. The PLOS ONE style templates can be found athttps://journals.plos.org/plosone/s/file?id=wjVg/PLOSOne_formatting_sample_main_body.pdf andhttps://journals.plos.org/plosone/s/file?id=ba62/PLOSOne_formatting_sample_title_authors_affiliations.pdf

2. Please update your submission to use the PLOS LaTeX template. The template and more information on our requirements for LaTeX submissions can be found at http://journals.plos.org/plosone/s/latex

Additional Editor Comments (if provided):

Two reviews are consistent with the quality of the manuscript, so I suggest the author make minor changes by taking the suggestions of the two reviewers into account.

Reviewers' comments:

Reviewer's Responses to Questions

**Comments to the Author**

1. If the authors have adequately addressed your comments raised in a previous round of review and you feel that this manuscript is now acceptable for publication, you may indicate that here to bypass the “Comments to the Author” section, enter your conflict of interest statement in the “Confidential to Editor” section, and submit your "Accept" recommendation.

Reviewer #1: (No Response)

Reviewer #2: All comments have been addressed

2. Is the manuscript technically sound, and do the data support the conclusions?

Reviewer #1: Yes

Reviewer #2: Yes

3. Has the statistical analysis been performed appropriately and rigorously? 

Reviewer #1: Yes

Reviewer #2: Yes

4. Have the authors made all data underlying the findings in their manuscript fully available?

Reviewer #1: Yes

Reviewer #2: Yes

5. Is the manuscript presented in an intelligible fashion and written in standard English?

Reviewer #1: Yes

Reviewer #2: Yes

6. Review Comments to the Author

Reviewer #1: I have not reviewed an earlier version of this paper.

Overall I think that it presents a fair assessment regarding the question of whether using past outbreaks to allow supervised learning has the potential to outperform unsupervised learning which is the main purpose of the work.

The conclusion "“….our results are promising that leveraging outbreak data with supervised learning will improve disease outbreak detection.” seems fair although the will might be softened to may.

I don't think that the approach as presented offers a reason to change to this for those currently deploying other algorithms - and this is not claimed by the authors. More generally, in common with the comparator approaches, the fact of an outbreak being probable is returned but this is a limited benefit. Although done a lot, this is very limited information and, for example, doesn’t make clear which cases belong to the outbreak and which do not. In practice the capacity of genetic sequencing and analysis to accurately detect and characterise outbreaks in Salmonella in particular makes such approaches largely redundant for this use case. The authors make a fair hand at noting that what is in practice detected is limited.

One technical issue is that modelling an outbreak as multiplicative relative to background rates appears strange. A secular trend of increasing incidence or a seasonal peak period would then need a larger outbreak to be detectable than when baseline levels are low. An additive model for the outbreak term might be a better fit. This approach was also applied to the simulation with outbreaks size proportional to the route of the variance of weekly counts such that the signal to be detected also carried this unlikely premise. The discussion might consider this choice in simulation and analysis and the alternative of non-multiplicative relationships.

The idea of seeking what is special about an outbreak vs modelling aberration from normal is appealing conceptually. I would have thought it might end up performing equivalently mathematically but the authors findings suggest that it does not and may be better in practice as well as a better theoretical fit as per their conclusion pasted in above.

Reviewer #2: Zacher and Czogiel propose a supervised HMM method for detecting potential disease outbreaks in Germany. This method takes advantage of the routinely collected outbreak data as known hidden states for improving detection performance. The effectiveness of this method was verified in experiments. The manuscript was well written. This paper will be ready for publication if the following minor problem is addressed.

Page 1, line 17

"on par or better than"  "on par with or better than"

7. PLOS authors have the option to publish the peer review history of their article (what does this mean?). If published, this will include your full peer review and any attached files.

Reviewer #1: No

Reviewer #2: No

---

## [Author Response · Author response to Decision Letter 1]

5 Apr 2022

Reviewer #1:

Overall, I think that it presents a fair assessment regarding the question of whether using past outbreaks to allow supervised learning has the potential to outperform unsupervised learning which is the main purpose of the work.

The conclusion “...our results are promising that leveraging outbreak data with supervised learning will improve disease outbreak detection.” seems fair although the will might be softened to may.

Response: We agree and changed it to “may”.

I don't think that the approach as presented offers a reason to change to this for those currently deploying other algorithms - and this is not claimed by the authors. More generally, in common with the comparator approaches, the fact of an outbreak being probable is returned but this is a limited benefit. Although done a lot, this is very limited information and, for example, doesn’t make clear which cases belong to the outbreak and which do not. In practice the capacity of genetic sequencing and analysis to accurately detect and characterise outbreaks in Salmonella in particular makes such approaches largely redundant for this use case. The authors make a fair hand at noting that what is in practice detected is limited.

Response: We agree that genetic sequencing is a great tool to characterize disease outbreak and overall the information our algorithm gives is limited. However, genomic Sequencing is not yet applied in an exhaustive manner. Genomic Sequencing might provide an accurate picture of an outbreak, but cases are also usually reported before genomic sequences are available and thus the surveillance of these time series might detect outbreaks earlier, which is important to prepare and implement public health measures in a timely manner.

One technical issue is that modelling an outbreak as multiplicative relative to background rates appears strange. A secular trend of increasing incidence or a seasonal peak period would then need a larger outbreak to be detectable than when baseline levels are low. An additive model for the outbreak term might be a better fit. This approach was also applied to the simulation with outbreaks size proportional to the route of the variance of weekly counts such that the signal to be detected also carried this unlikely premise. The discussion might consider this choice in simulation and analysis and the alternative of non-multiplicative relationships.

Response: In the time series of infectious disease case counts, the variance usually increases with the background rates and thus the detection of small outbreaks becomes more difficult with increasing background, also when using an additive model. However, if the background rates are high, the relevance of small outbreak for public health surveillance and infection control is limited, because small outbreaks will not contribute much to the overall occurrence of infection. Nonetheless, we agree that this is a limitation of our model and we now mention this in the discussion.

The idea of seeking what is special about an outbreak vs modelling aberration from normal is appealing conceptually. I would have thought it might end up performing equivalently mathematically but the authors findings suggest that it does not and may be better in practice as well as a better theoretical fit as per their conclusion pasted in above.

Reviewer #2:

Zacher and Czogiel propose a supervised HMM method for detecting potential disease outbreaks in Germany. This method takes advantage of the routinely collected outbreak data as known hidden states for improving detection performance. The effectiveness of this method was verified in experiments. The manuscript was well written. This paper will be ready for publication if the following minor problem is addressed.

Page 1, line 17: "on par or better than"  "on par with or better than"

Response: Changed.

---

## [Editor Report · Decision Letter 2]

11 Apr 2022

Supervised learning using routine surveillance data improves outbreak detection of Salmonella and Campylobacter infections in Germany

PONE-D-21-39547R2

Dear Dr. Zacher,

We’re pleased to inform you that your manuscript has been judged scientifically suitable for publication and will be formally accepted for publication once it meets all outstanding technical requirements.

Kind regards,

Hong Qin

Academic Editor

PLOS ONE

---

## [Editor Report · Acceptance letter]

26 Apr 2022

PONE-D-21-39547R2 

Supervised learning using routine surveillance data improves outbreak detection of Salmonella and Campylobacter infections in Germany 

Dear Dr. Zacher:

I'm pleased to inform you that your manuscript has been deemed suitable for publication in PLOS ONE. Congratulations! Your manuscript is now with our production department. 

Kind regards, 

on behalf of

Dr. Hong Qin 

Academic Editor

PLOS ONE